

# Covariance-informed spatiotemporal clustering improves the detection of hazardous weather events

Hunter Quintal[1], Antonia Sebastian[1], Marc Serre[2], Wiebke Jäger[3], Marleen C. de Ruiter[3]

[1]Department of Earth, Marine, and Environmental Sciences, The University of North Carolina at Chapel Hill, Chapel Hill, 27599, USA
[2]Department of Environmental Sciences and Engineering, The University of North Carolina at Chapel Hill, Chapel Hill, 27599, USA
[3]Institute for Environmental Studies, Vrije Universiteit Amsterdam, Amsterdam, 1081 HV, the Netherlands

*Correspondence to*: Hunter Quintal (hquintal@email.unc.edu)

**Abstract.** Spatiotemporal clustering can be used to detect weather events in multi-dimensional datasets. This method requires that the resolution of a dataset equivalently resolves fluctuations across space and time, thereby normalizing the dataset for unbiased clustering across three dimensions. Yet, few studies test whether a dataset meets this requirement as there is no standard approach to do so. To address this methodological gap, we present a framework to quantify the relationship between space and time using space time separable covariance modelling. We demonstrate that, by defining a temporal resolution of interest (e.g. hours, days), the equivalent spatial resolution can be empirically derived using a space time metric. We present an application using the unsupervised machine learning method Density-Based Spatial Clustering of Applications with Noise (DBSCAN) to detect heat waves and severe storms across the Southeastern US from 1940 to 2023 from ECMWF Reanalysis version 5 (ERA5) data. We analyse the seasonal behaviour of space time metrics for precipitation and heat index before selecting representative values. We find that both ERA5-derived daily heat index and hourly precipitation are insufficiently resolved for unbiased clustering at their native resolutions (i.e., 0.25 spatial degrees [degree] per day for heat index and 0.25 degree per hour for precipitation). We show that a resolution of 0.39 degree per day (0.05 degree per hour) prevents preferential clustering in either the spatial or temporal dimension for heat index (precipitation). We hypothesize that event identification will improve by resampling the data by the space time metric. Heat wave clusters that were produced using the unbiased resolution were compared against the NOAA Storm Events Database from 2019 to 2023. Recall of heat waves increased from 0.92 to 0.94 using the covariance-informed resolution, demonstrating the importance of normalization prior to weather event reconstruction. Ultimately, the inclusion of temporal geostatistics leads to improved reconstruction of historical weather events and enables evaluation of their scale and variability.

**Plain-text summary.** High quality weather event datasets are crucial to community preparedness and resilience. Researchers create such datasets using clustering methods, which we advance by addressing current limitation in the relationship between space and time. We propose a method to determine the appropriate factor by which to resample the spatial resolution of the data prior to clustering. Ultimately, our approach increases the ability to detect historic heatwaves over current methods.



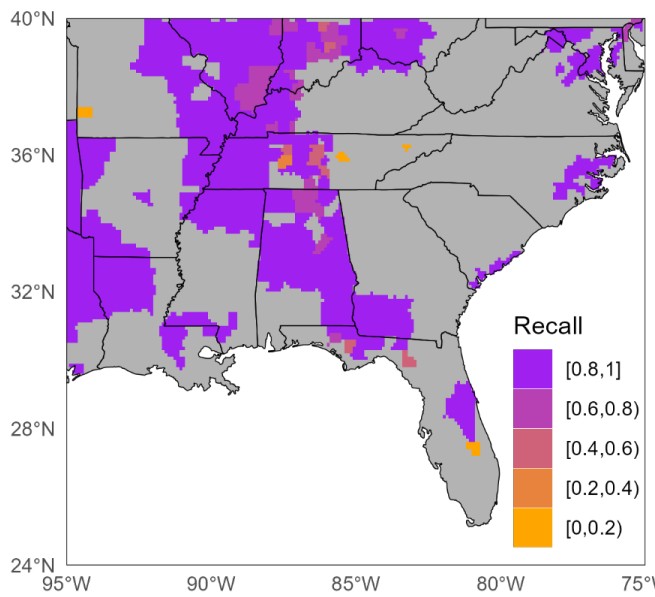

**Key Figure. Recall of NOAA excess heat episodes (county scale) using covariance-informed spatiotemporal clustering of heat index (2019-2023). A value of 1.0 indicates that the model recalls all the weather events reported for that county.**
**Counties coloured grey never reported excess heat during this period. Aggregate recall = 0.94.**

### 1.   Introduction

Understanding trends in extreme weather events is crucial to preparedness and adaptation (Ward et al., 2020). However, incomplete historical reporting of weather events has prevented comprehensive understanding of their intensity, duration, extent, and frequency (Paprotny et al., 2018, 2024). Advances in geophysical event detection within observational, reanalysis, and remotely sensed datasets has grown in recent years, leading to novel insights into event characteristics (Yu et al., 2020). However, differences in spatiotemporal coverage, resolution, data quality, measurement, and detection methods have resulted in inconsistent weather event definitions. This has been shown to impact hazard modelling results, leading to unreliable risk assessments and undermining risk management (Agulles et al., 2024). Extreme weather events are often identified using globally resolved reanalysis data to reduce the limitations discussed above and improve consistency in weather event descriptions. At the regional to global scale, the application of spatiotemporal clustering to European Centre for Medium-Range Weather Forecasts (ECMWF) Reanalysis v5 (ERA5) meteorological datasets is now widely practiced for tracking time-evolving weather events including storms, drought, and heat waves (Barton et al., 2016, 2022; Dey & Chakraborty, 2015; Liu et al., 2023; Liu & Zhou, 2023a; M. Luo et al., 2024; M. Luo, Wang, et al., 2022; Rogers et al., 2022; Tang et al., 2019; Tilloy et al., 2022; Tin Tin et al., 2024; Wibisono et al., 2021).

Spatiotemporal clustering is an unsupervised machine learning method regularly applied to weather event detection that groups

extreme observations by their proximity in space and time (Ansari et al., 2020). This method assumes that spatial and temporal

distances be equal prior to clustering, enabling the search for neighbouring extreme observations to be equally plausible across

space and time and thus preventing preferential clustering in either dimension (Birant & Kut, 2007). To meet the normalization

definition, the dataset is defined over a grid made of elemental space time cubes whereby the variation along the spatial dimension

of the cube is equivalent to the variation along its temporal dimension (Bach et al., 2017). However, researchers often assume that

the native resolution of a dataset satisfies this normalization definition without formal testing, thus treating time as simply another

spatial dimension. This decision is problematic when the native spatial and temporal dimensions of the data do not match that of

the physical processes that control hazardous weather events, leading to missed event detection (e.g., convective storms may only

appear in datasets whose resolution resolves convective processes). A test is therefore needed to determine whether the resolution

of a three-dimensional dataset meets the definition of a space time cube, which may also inform how to resample a dataset into a

60 space time cube when the normalization definition is not met.

Geostatistical space/time covariance modelling (Kolovos et al., 2004) provides a natural framework within which to study the

spatial and temporal distance scales of the variability of the physical processes that control them. It also enables researchers to test

the requirement that space time cubes are properly scaled prior to clustering, yet it has not been widely applied to weather event

data. We hypothesize that the inclusion of covariance modelling into the spatiotemporal clustering framework would enable testing

of bias in the space time resolution, alter the weather events identified, and improve the identification of weather events in the

historical record. In this paper, we introduce a method for quantifying a 'space time metric' that can be used to establish whether

the space time resolution of a three-dimensional dataset corresponds to a 'space time cube' (Kisilevieh et al., 2010). It also

quantifies the fluctuation of a variable in space relative to time at the resolution of a given dataset. We interpret a fluctuation to

represent the scale (extent and duration) of a corresponding weather event. As such, seasonal variability in the physical processes

leading to weather events may subsequently affect space time metrics. We demonstrate the value of using that space time metric

to test whether the resolution of a dataset meets the definition of a properly normalized space time cube prior to clustering. To

establish the utility of the proposed covariance-informed spatiotemporal clustering method, we apply our approach to ERA5-

derived heat index and precipitation estimates to identify historical heat waves and severe storms and find that clustering at the

native resolution leads to the identification of different weather events that are less representative of those reported in the historical

record than when clustering at the covariance-informed resolution. We also draw inferences about possible atmospheric and

climatological processes controlling the seasonal behaviour of space time metrics.



While the geostatistical tools for deriving a space time metric are not new, their application to spatiotemporal clustering is. Without normalization, the spatiotemporal extent of a given weather event may be partially missed or exaggerated, leading to misinformed hazard, exposure, and vulnerability analyses. We resolve this issue by using covariance to resize the spatial resolution to match the temporal resolution. The novelty of this study is threefold. First, this study is the first application of space time modelling to properly size weather events such as heat waves and severe storms. Second, by enabling autofitting of experimental covariance, this study will be one of the first assessments of whether single or nested covariance models are better for capturing the structure of climate fluctuations. Finally, this study is the first application of covariance modelling to create space time metrics for the purpose of unbiased spatiotemporal clustering.

## 1.1 Related works and basic concepts

Multiple meteorological datasets exist for weather event detection. Datasets are assembled as observational, remotely sensed, modelled, reanalysis, and reported products. The quality of each dataset varies with inherent assumptions, biases, and resolutions, each of which impacts the ability to detect the extent of a weather event (Hassler & Lauer, 2021; Polasky et al., 2025). Furthermore, trade-offs exist with each dataset. For example, gauge observations offer temporal continuity but are spatially discontinuous (Kidd et al., 2017). Whereas remotely sensed datasets provide spatially continuous measurements but are temporally discontinuous due to time between satellite overpasses (Alvera-Azcárate et al., 2007). Meanwhile, radar-rainfall products offer both spatial and temporal continuity but are limited to precipitation estimates in data-rich regions and have a short record (e.g., Next Generation Radar (NEXRAD) has been operational since 2002) (Saltikoff et al., 2019). Increasingly, event detection is conducted using modelled and reanalysis products, which also maintain spatiotemporal continuity at global scale but are biased by aleatory and epistemic uncertainties and have coarse spatial resolution with respect to the scale of weather events (Steinschneider et al., 2015).

Interest in application of spatiotemporal clustering methods to detect weather events is growing (Dey & Chakraborty, 2015; Pamuji & Rongtao, 2020; Tin Tin et al., 2024; Wibisono et al., 2021). Spatiotemporal clustering algorithms employ unique search strategies when mining geospatial datasets, facilitating evaluation of specific research aims. These strategies can be broadly distinguished as those useful for discovering a pre-defined number of clusters (partitional, e.g. k-means), relational clusters within a nested tree structure (hierarchical), and irregularly shaped clusters (density-based, e.g. Density-Based Spatial Clustering of Applications with Noise (DBSCAN)) (Reynolds et al., 2006). These methods are most often applied to univariate three-dimensional datasets, and while some multivariate applications exist, most multivariate clustering is conducted with spectral techniques to reduce complexity associated with high dimensionality (Choi & Hong, 2021; L. Wang & Wang, 2025; R. Xu & Wunsch, 2005). In this study, we evaluate and apply univariate clustering methods to produce distinct weather event datasets.

While multiple clustering algorithms have been applied for weather event detection, many ignore evaluating whether the dataset is normalized prior to clustering, leaving potential biases unknown. For example, applications using the spatiotemporal connectivity algorithm (Yin et al., 2025), the 3D connected component algorithm (CC3D) (Liu & Zhou, 2023b; M. Luo, Lau, et al., 2022), and DBSCAN (Tilloy et al., 2022) all neglect to account for normalization, which may result in biased weather event detection and subsequent event characteristics (Birant & Kut, 2007). Other applications include geo-referenced data item

clustering, moving clusters, trajectory clustering, and semantic-based trajectory data mining (Ansari et al., 2020).

Of all the aforementioned methods, DBSCAN holds promise for weather event detection because of its fast implementation and ability to detect clusters of irregular size without a priori knowledge of the number of clusters, enabling efficient evaluation of time-evolving weather events across space and time. DBSCAN requires a threshold to identify extreme points and a point density to define the fewest points within a neighbourhood search radius needed to form a cluster (Fig. 1), where the smallest possible

point density must exceed the number of dimensions of the dataset (Ester et al., 1996). Spatiotemporal (ST)-DBSCAN further assumes that spatial and temporal dimensions are equivalent to reduce bias when searching for neighbouring points (Birant & Kut, 2007). The neighbourhood search radius is calculated using $k^{th}$ nearest neighbour (kNN) analysis, where k is the user-defined point density. When the neighbourhood search is made across space *and time*, the spatial and temporal dimensions of the dataset must be equivalent to ensure that the search will identify clusters of space time cubes that do not bias either space or time.

A growing number of applications do consider normalizing spatial and temporal distances prior to spatiotemporal clustering, including density-based algorithms such as ST-DBSCAN (Birant & Kut, 2007), ST-GRID (M. Wang et al., 2006), ST-OPTICS (Agrawal et al., 2016; Andrienko & Andrienko, 2010), SNN-4D+ (Oliveira et al., 2013), and CorClustST (Hüsch et al., 2020) as well as hierarchical algorithms like ST-HC (Lamb et al., 2020) and ST-HDBSCAN (Rus et al., 2022). Each algorithm employs a unique approach to normalize spatial and temporal distances. Some search for neighbouring points to cluster in space before

searching in time (e.g., ST-DBSCAN, ST-GRID). Others combine space and time into one spatiotemporal distance prior to clustering, either by weighting space and time distances (e.g., SNN-4D+), weighting space, time, and attribute distances (e.g., ST-HC), considering spatial correlations over time (e.g., CorClustST), or proportionally transforming temporal distances into equivalent spatial distances (e.g., ST-OPTICS). However, no existing algorithm enables researchers to define a relevant weather event duration prior to clustering that may vary from sub-hourly mesocyclones (e.g., Li et al., 2008a; 2008b) to seasonal droughts

(e.g., Cammalleri & Toreti, 2023; Liu & Zhou, 2023b) by proportionally transforming spatial distances into equivalent temporal distances. In the absence of methods that resolve these practical requirements, techniques are needed to fill a two-fold methodological gap: the ability to test whether the spatial and temporal dimensions of a dataset are equivalent and, if they do not,



a method by which a space time metric (i.e., normalization factor) can be used to convert the data into a space time cube to reduce bias in subsequent results.

## 2. Proposed framework

### 2.1 Covariance modelling

We introduce a covariance-informed space time metric that can be used to adjust the space time resolution of a dataset so that it meets the requirement needed to perform spatiotemporal clustering using DBSCAN (Figure 1). We formally describe this requirement as follows: the space time resolution of a dataset is the spatial distance and time increment of the elemental space time

grid cell of that dataset. A space time cube is then some elemental space time grid cell with a distance along the spatial dimension that experiences the same variation as the variation along its temporal dimension. A dataset meets the requirement needed for spatiotemporal clustering if its space time resolution is a space time cube. Here, we present a geostatistical method to establish the unbiased space time resolution of a dataset and then demonstrate how this method can be used to derive a covariance-informed space time metric within the spatiotemporal clustering framework. This geostatistical methodology separately quantifies

experimental covariance as a function of spatial lag and temporal lag using space time separable covariance analyses developed in the Bayesian Maximum Entropy (BME) method of modern spatiotemporal geostatistics and its numerical implementation (Christakos et al., 2002; Serre & Bogaert, n.d.; Serre & Christakos, 1999). Statistical models are fit to the experimental covariance, from which a space time metric can be empirically derived.

We hypothesize that the space time metric would enable the modeler to quantify the relationship between spatial and temporal

distances and thus resample the spatial resolution such that the resampled dataset resolution meets the definition of a space time cube while preserving the temporal resolution. Space time separable covariance modelling has been applied to numerous geophysical studies, including groundwater quality (Gómez et al., 2021), surface water quality (Holcomb et al., 2018), air pollution (Delang et al., 2021), and disease and fatalities (Fox et al., 2015). Therefore, while physical space time metrics have been used to define the distance between points in the space time domain for environmental modelling applications (e.g., temperature as in

Christakos et al. (2017)), to our knowledge, this is the first application of space-time separable covariance modelling to identify extreme weather events within a spatiotemporal clustering framework. We leverage the BMElib package, which implements BME theory for both spatial and space time geostatistics, dealing with both continuous and categorical variables, implementing the best unbiased nonlinear estimator (BME), which rigorously processes hard and soft data (Christakos & Serre, 1999). BMElib also leads to the (simple, ordinary, universal) kriging methods as linear limiting cases (Christakos & Li, 1998). The package employs the

concepts of space time metrics, space time neighbourhood search, space time covariance analysis, and space time estimation because the BMElib package was designed for space time geostatistics. As a result, BMElib provides better geostatistical functions



for space time analysis than classical geostatistical software (where time is included merely as another spatial dimension) (Christakos, 1990). Many of the studies described in Sect. 1.1 follow this classical geostatistical assumption.

In this section, we advance the current application of DBSCAN for spatial clustering described on row 1 of Fig. 1. Our application

first incorporates space time separable covariance modelling of a gridded time series of observations to model the behaviour of a given variable separately as functions of spatial and temporal distance (Fig. 1, row 2). This technique enables quantification of the covariance of a variable in space and in time, therefore allowing the relationship between the two dimensions to be evaluated as the ratio between the fluctuation in space and the fluctuation in time. This ratio, termed the space time metric, characterizes the rate of fluctuation in either dimension. We describe the geostatistical theory behind space time separable covariance modelling and

present equations needed for the derivation of space time metrics from the covariance model. We then interpret that the resolution of the dataset equivalently resolves fluctuations in space and time if the space time metric is equal to one. A space time metric that is equal to one results in unbiased clustering, greater than one leads to preferential clustering in time (requires spatial coarsening) and less than one leads to preferential clustering in space (requires spatial interpolation). Furthermore, we propose to resample the dataset if the metric is greater than one to reduce bias during clustering. Upon evaluation of the space time metric, and subsequent

resampling, application of DBSCAN for spatial clustering can resume as normal, albeit with a resampled space time cube rather than the raw dataset (Fig. 1, row 2, column 2).



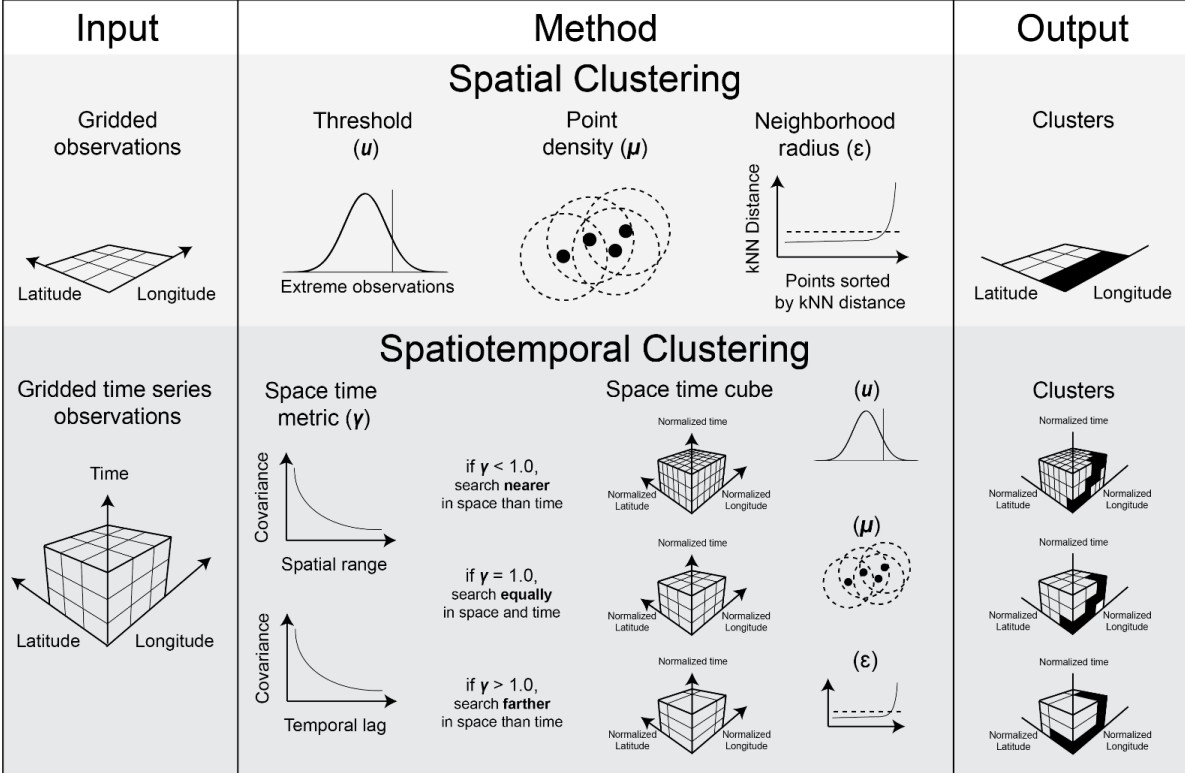

**Figure 1: Approach to clustering using Density-Based Spatial Clustering of Applications with Noise (DBSCAN). Clustering in space only (light grey row) groups together a minimum number ($\mu$) of extreme points ($u$) required within a neighborhood radius ($\varepsilon$). Clustering in space and time (dark grey row) requires the dataset be formatted as a space time cube whereby spatial distances are made equivalent to temporal distances prior to clustering. A space time cube with equivalent space and time dimensions can be created by deriving a space time metric ($\gamma$) using space time separable covariance analysis and resampling the native resolution of the data by $\gamma$. Similar to spatial clustering, $u$, $\mu$, and $\varepsilon$ can then be defined and used to identify clusters of extreme points.**

We let $X(\boldsymbol{s},t)$ be a Space/Time Random Field (S/TRF) representing the weather variable of interest at spatial coordinate $\boldsymbol{s}$ and time $t$, and $c_X((\boldsymbol{s},t),(\boldsymbol{s'},t')) = E[(X(\boldsymbol{s},t) - m_X(\boldsymbol{s},t))(X(\boldsymbol{s'},t') - m_X(\boldsymbol{s'},t'))]$ be its space/time covariance, where $m_X(\boldsymbol{s},t) = E[X(\boldsymbol{s},t)]$ is its mean and E[.] is the expectation operator. For homogenous S/TRFs the covariance is homoscedastic and only a function of the spatial lag $r = \|\boldsymbol{s} - \boldsymbol{s'}\|$ and time difference $\tau = |t - t'|$, i.e. $c_X((\boldsymbol{s},t),(\boldsymbol{s'},t')) = c_X(r,\tau)$ (Christakos et al., 2000). The covariance function can be modelled as a space/time separable Gaussian model, $c_X(r,\tau) = \sigma_X^2 exp \frac{-r^2}{3a_r^2} exp \frac{-\tau^2}{3a_\tau^2}$ or exponential model $c_X(r,\tau) = \sigma_X^2 exp \frac{-r}{3a_r} exp \frac{-\tau}{3a_\tau}$, where $\sigma_X^2$, $a_r$ and $a_t$ are the variance, the spatial range, and the temporal range, respectively, representing the variability in weather parameter values, the spatial distance over which weather fluctuations spread geographically, and the temporal duration of these fluctuations. In the more general case, there may be nested fluctuations, which can be modelled as the sum of space/time separable covariance structures



$$c_x(r,\tau) = \sigma_X{}^2 \left[ \alpha_1 exp \frac{-r^2}{3a_{r1}{}^2} exp \frac{-\tau^2}{3a_{\tau1}{}^2} + \alpha_2 exp \frac{-r}{3a_{r2}} exp \frac{-\tau}{3a_{\tau2}} + \cdots \right]$$
(1)

where $\alpha_1$, $\alpha_2$, etc. are the proportion of the overall variance $\sigma_X{}^2$ expressed by each covariance structure, $a_{r1}$ and $a_{t1}$ are the spatial

and temporal dimension of fluctuations represented by the first structure, $a_{r2}$ and $a_{t2}$ are those of the second structure, and so on.

As can be seen from this framework, covariance models are naturally suited to quantify the spatial and temporal dimensions of weather fluctuations, which we use in the next section to define the space time metric.

**2.2 Space time metric**

Space time separable covariance models can be fit to the experimental covariance values and optimized by minimizing the sum of

squared error (SSE). A covariance-informed space time metric is empirically derived using an optimal covariance model that is

physically representative of the persistence of the fluctuation of a given variable across space and time. The space time metric can

be calculated in three ways using the optimized space time separable covariance models described above. In the case of a single

structured covariance model, the space time metric ($\gamma$) is calculated using Eq. (2) as the ratio of spatial range ($a_r$) to temporal

range ($a_\tau$). This simple ratio directly assesses the spatial extent of a fluctuation relative to its duration, which is defined by the

space time separable covariance models. This approach has never been used before in the context of spatiotemporal clustering.

$$\gamma_1 = \frac{a_r}{a_\tau}$$
(2)

In the case of a nested covariance model containing multiple structures, the space time metric can be calculated in two ways to

resolve the complexity of each additional covariance structure. In the first instance, the space time metric can be calculated using

Eq. (3) as a ratio of the variance weighted average of spatial ranges over the variance-weighted average of temporal ranges.

$$\gamma_2 = \frac{(\alpha_1 * a_{r1} + \alpha_2 * a_{r2} + \cdots)}{(\alpha_1 * a_{\tau1} + \alpha_2 * a_{\tau2} + \cdots)}$$
(3)

Alternatively, $\gamma$ can be calculated using Eq. (4) as the variance-weighted average of ratios of spatial ranges over temporal lags for

each covariance structure.

$$\gamma_3 = \alpha_1 * \frac{a_{r1}}{a_{\tau1}} + \alpha_2 * \frac{a_{r2}}{a_{\tau2}} + \cdots$$
(4)



### 2.3 Hypothesis testing

Our purpose is to define a space time metric that can be used to test whether the space time resolution of a dataset represents an unbiased space time cube (Fig. 1, row 2, column 2). To do this, we set the spatial unit to the spatial resolution and the temporal unit to the temporal resolution. For example, if the dataset is provided at a resolution of 0.25 degree by 1 hour, then one spatial

unit equals 0.25 degree, one time unit equals 1 hour, and the space time metric of Eq. (2) to Eq. (4) provide the ratio of the geographical extent of weather fluctuations expressed by increments of 0.25 degree divided by their duration expressed in increments of 1 hour. Hypothesis testing can be conducted following the calculation of such a space time metric. The null hypothesis ($H_0$) to test is that the space time metric equals one, implying that spatial distances (numerator) are equivalent to temporal distances (denominator) for the given dataset. The alternative hypothesis ($H_A$) is that the space time metric does not equal

one, which we infer to mean that spatial distances are not equivalent to temporal distances.

$$H_0: space\ time\ metric = 1\ ;\ H_A: space\ time\ metric \neq 1$$

The given dataset satisfies the assumption of spatiotemporal clustering if the null hypothesis cannot be rejected. In this case, the resolution of the dataset would meet the definition of a space time cube and spatiotemporal clustering can be conducted as usual. If the null hypothesis is rejected, then the resolution of the dataset would not satisfy the definition of a space time cube.

Furthermore, if the null hypothesis is rejected, we would interpret the space time metric to be an empirical factor by which to spatially resample a given dataset so that it would meet the definition of a space time cube. A statistical test must first be selected to evaluate the null hypothesis. A parametric test, such as a one-sample t-test, may be considered if the dataset displays both normality and independence. If these parametric assumptions are not met, a nonparametric test may be considered such as the Wilcoxon signed-rank test. We apply the Shapiro-Wilk test to determine whether each dataset is normally distributed before

selecting an appropriate test (Davis, 2002).

3.   **This interpretation enables the creation of a space time cube from any dataset, whereby the space time metric is treated as an empirical factor by which to resample the native spatial resolution. A space time metric greater than one would indicate spatial coarsening is needed while a metric less than one would indicate spatial interpolation. In practice, spatial interpolation is not feasible because no new information is added; in this case a space time**

**metric less than one would instead be multiplied by the native spatial resolution to define the unbiased resolution of a dataset needed to satisfy the definition of a space time cube. Upon resampling, unbiased spatiotemporal clustering is possible using DBSCAN or any clustering algorithm. Application**

### 3.1 Study region

The Southeastern United States (SEUS) is at a crossroads of risk. The region (Fig. 2) is frequently exposed to natural hazards

including hurricanes, tropical storms, tornadoes, coastal flooding, inland flooding, drought, and wildfire (Iglesias et al., 2021; Summers et al., 2022) and these events are expected to change in frequency and severity under a changing climate (Reidmiller et



al., 2018). The SEUS is also home to many sprawling urban centres, whose urbanized areas are predicted to double or triple over the next forty years, further exposing high populations and infrastructure to severe weather events (KC et al., 2021; Terando et al., 2014). Across the United States, rural southeastern populations are the most frequent hotspots of both high social and financial vulnerability to natural hazards, indicated by high application rates for short-term post-disaster assistance (Drakes et al., 2021). These contributing risk factors underscore the need for high quality weather event datasets in this region to improve preparedness and adaptation under a changing climate.

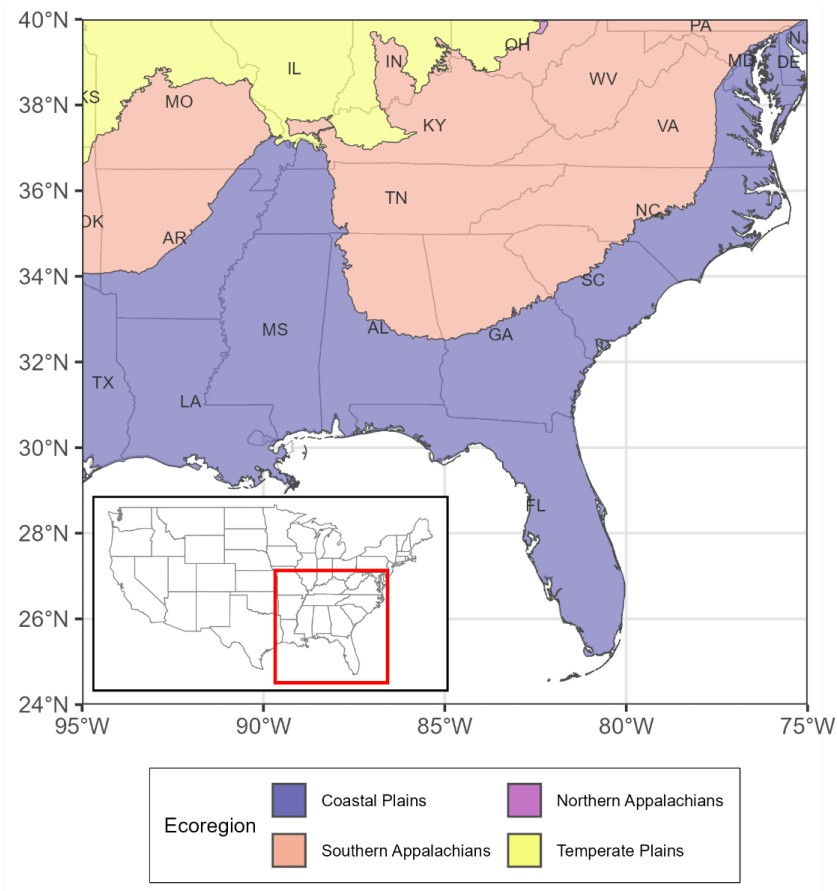

**Figure 2: The Southeastern United States, outlined in red, is considered in this analysis. This region comprises Coastal Plains, Appalachian Mountains, and Temperate Plains landscapes.**

**3.2 Data collection**

Meteorological variables including total precipitation, temperature, and dewpoint temperature were obtained from ERA5 from 1940 to 2023; a global, hourly resolved climate reanalysis dataset two meters above surface along a 0.25 degree grid (Hersbach et al., 2020). ERA5-derived temperature estimates are popular when quantifying the spatiotemporal extent of heat waves using

spatiotemporal clustering (e.g., (Chen et al., 2021; C. Li et al., 2022; Liu et al., 2023; M. Luo, Lau, et al., 2022; M. Luo, Wang, et al., 2022)). We consider heat index, rather than temperature, to detect extreme humid heat potentially hazardous to human health (Di Napoli et al., 2019; Horton et al., 2016; Raymond et al., 2021, 2022; Rogers et al., 2021). This decision is due to the growing field of evidence of the impacts associated with exposure to humid heat (Baldwin et al., 2023; D. Li et al., 2020; Raymond et al.,
2020).

Hourly heat index was calculated from temperature and dewpoint temperature using the weathermetrics R package following the approach used by NOAA's National Weather Service (NWS) (Anderson et al., 2013). For more information concerning heat index, the reader is referred to Spangler et al. (2022). Not Available (NA) values result when the heat index equation produces physically implausible solutions. Therefore, NA values were assigned the focal mean using a window of nine neighbouring cells within the
same timestep. Interpolation across timesteps gives heat index estimates to a given latitude and longitude where remaining NAs exist. Remaining NA values at the beginning or end of the time series were filled using a linear regression with the two nearest raster timesteps with observations at that spatial location. Heat index was then summarized as the maximum daily value to identify weather events that span multiple days, which may otherwise be missed.

Previous studies have also applied spatiotemporal clustering to ERA5-derived total precipitation estimates (Liu & Zhou, 2023a;
Tilloy et al., 2022). In this study, we calculate the 24-hour precipitation volume in each hour using a rolling sum because this accumulation window is frequently used as input for flood modelling analyses (Barbero et al., 2019). Unlike heat index, precipitation was not summarized at a daily temporal resolution because extreme precipitation is unlikely to persist across multiple days, except in the most severe storms. This choice of temporal resolution enables detection of smaller and shorter storms than would be observable at a daily temporal resolution.

Thresholds above which observations were considered extreme were then created using two National Oceanic and Atmospheric Administration (NOAA) datasets (Fig. 3). Heat index thresholds were downloaded for NOAA Weather Forecast Office (WFO) County Warning Areas (CWA) (Allen, 2024) from NWS Directive 010 (Cooper, 2018; Strager, 2019; Tuell, 2017). Precipitation thresholds were based on NOAA Atlas 14 Annual Recurrence Intervals (ARI) 24-hour 1-year return period volumes resolved at 0.0083 spatial degrees, downloaded from NWS's Precipitation Frequency Data Server (Bonnin et al., 2006; Perica, Deborah, et
al., 2013; Perica, Martin, et al., 2013). We evaluated extreme observations about the 1-year return period to obtain sufficient observations for clustering that may still pose a pluvial hazard to exposed regions.



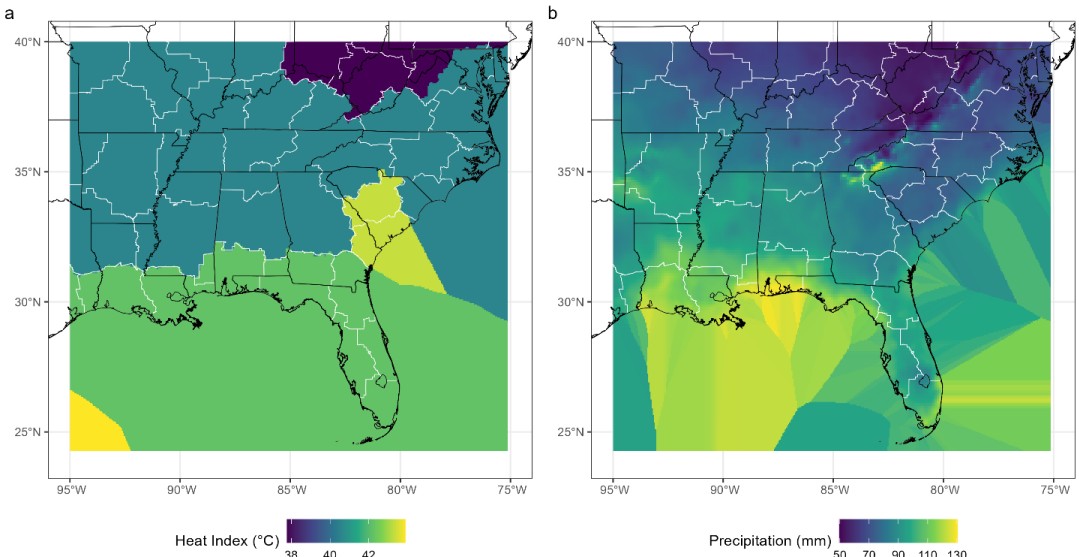

**Figure 3. a) Heat index and b) precipitation thresholds quantifying extreme observations across the domain. Heat index values include those required to issue a heat advisory alert in each NOAA NWS Weather Forecast Office's (WFOs) County Warning Area (CWA) outlined in white. Thresholds were then expanded over water using nearest neighbour values. Precipitation values include those exceeding the 24-hour accumulated 1-year return period precipitation estimated by NOAA Atlas 14 Annual Recurrence Intervals.**

### 3.3 Covariance modelling

The heat index and precipitation datasets were transferred into MATLAB to quantify experimental covariance in both space and time aggregated by month using the BMElib package (Christakos et al., 2002; Kolovos et al., 2004; Serre & Christakos, 1999). Experimental covariance was calculated in monthly chunks for daily-resolved heat index and for hourly-resolved precipitation. The BMElib processing time is reported in Table S2.

Four different space time separable covariance models were then fit to the experimental covariance for each month from 1940 to 2023. These models include a single gaussian, a single exponential, a nested exponential-exponential, and a nested gaussian-exponential. Each model was then optimized by minimizing the SSE between the model and the experimental covariance in each month. Furthermore, in the case of the nested covariance models, this optimization was then recursively conducted for all possible $\alpha$ values from zero to one hundred percent by a one percentage increment until a single optimal model was selected for that specific month. Then, the model which minimized SSE across the four possible covariance models was selected to calculate the space time metric for each month. Covariance-informed space time metrics were calculated by month for daily-resolved heat index and hourly-resolved precipitation observations using Eq. (3) and Eq. (4) from 1940 to 2023. We then conducted statistical testing to evaluate the null hypothesis. We sought to quantify whether the median of the space time metric equals one, which would determine whether spatial distances are equivalent to temporal distances within the ERA5 dataset.



We define the dataset as a space time cube if the null hypothesis cannot be rejected. If the null hypothesis is rejected, we create a space time cube by spatially resampling the dataset by a factor equivalent to the space time metric. Since severe storms may plausibly occur at any time of year, the record median of space time metrics calculated each month is sufficient for resampling

ERA5-derived precipitation into a space time cube. However, since heatwaves primarily occur in the summer but may occur across the remaining seasons, we evaluate heat index clusters resampled using the summer and the record median to test the sensitivity of clustered heatwaves to the space time metric.

### 3.4 Clustering

#### 3.4.1 Parameters

Meteorological observations within a space time cube were compared against a threshold to define extreme observations subject to spatiotemporal clustering. Extreme observations would then represent conditions favourable either for the alert of a heat advisory in the case of heat index or for an annually occurring storm in the case of precipitation. The extreme observations produced with thresholding were then clustered. In this study, the smallest detectible heatwave or storm event using DBSCAN spans $\mu = 4$ extreme observations across space and/or time that all reside within the neighbor radius ($\varepsilon$) determined by $k^{th}$ nearest neighbour

(kNN) analysis. The neighbour radius is defined using kNN analysis developed in the pathviewr R package (W. Luo & Brouwer, 2013), whereby all extreme points are sorted by their normalized distance in space time to their $k = \mu - 1$ nearest neighboring extreme observations. Then, the elbow method is applied to the points, sorted by the $3^{rd}$ nearest neighbour distance, to identify extreme observations to be included in a cluster from those that will be discarded as isolated noise.

#### 3.4.2 Clustering with DBSCAN

Spatiotemporal clustering was then performed against the latitude, longitude, and date time entries for each dataset using the dbscan R package (Hahsler et al., 2019). First, the three dimensions were standardized such that the lowest values of latitude, longitude, and date time were each represented by a value of one. Then, each sequentially larger value of each dimension was assigned an integer value one larger than the previous until a three-dimensional series of integers represented the dataset. Then, kNN analysis was conducted on the normalized latitude, longitude, and date time entries for $\mu = 4$ to quantify the value of $\varepsilon$ for each dataset.

Finally, spatiotemporal clustering was completed for the relevant $\mu$ and $\varepsilon$ for each dataset. The inputs and outputs of spatiotemporal clustering are reported in Table 1.

### 3.5 Validation

We evaluate the model performance of clustered weather events reported by professional meteorologists to the NOAA Storm Events Database, which is one of the few databases to report impactful weather events from both precipitation- and heat-based

natural hazards across CONUS. The NOAA Storm Events Database was obtained through a bulk data download from the NOAA

National Centers for Environmental Information (NCEI) (Konisky et al., 2016). The database, which includes weather event

information from January 1950 to November 2024 as reported by NWS, reports storm events and episodes of sufficient intensity

to cause mortality, morbidity, or financial loss, disruption to commerce, rare or unusual weather phenomena, and other significant

weather events. Storm episodes contain multiple storm events, each with a unique episode and weather event identifier, start and

end dates and times, a location, and impacts including financial losses, injuries, and fatalities. Locations are reported as the exposed

county and, for some hazard episodes, by the centroid of the exposed area. NOAA storm episodes were subset to those determined

to be associated with a heavy storm (flash flood, flood, heavy rain, hurricane, tropical depression, tropical storm, and typhoon) or

a heat wave (heat and excessive heat). These files were rasterized to the county resolution to ensure consistency in exposure units.

A summary of these NOAA events is included in Table S1.

Clusters were then compared to NOAA storm episodes to quantify model performance under the native resolution of ERA5 and

the covariance-informed space time cubes. Model performance was determined by calculating a confusion matrix. In this analysis,

we treat the NOAA Storm Events Database as the observed dataset and the clustered weather events as the modelled dataset. In

selecting a performance metric for evaluation, we must consider that both observed and modelled datasets are unbalanced: many

more days exist without than with hazardous conditions. Furthermore, we expect many more clusters than have been reported to

NOAA, which suggests the need for a metric that does not penalize false positives in the confusion matrix. Therefore, we calculate

recall for model evaluation because this metric is designed to work well with unbalanced datasets and does not penalize false

positives. Recall tests how well a modelled dataset predict an observed dataset, where a value of 0 indicates that the model never

identifies the observation, and a value of 1 indicates that the model always identifies the observation.

Clusters were resampled to overlapping counties to match the resolution of the NOAA Storm Events Database. Confusion matrices

were then developed to compare how well clustered weather events detect storms in the NOAA Storm Events Database. Confusion

matrices were calculated two ways: across the domain per day to quantify aggregated recall and within each county to quantify

county-specific recall across the comparison period. To ensure the highest quality of NOAA data for comparison, we only assess

performance against NOAA Storm Events reported between 2019 and 2023.





## 4. Results

### 4.1 Covariance modelling

Figure 4 depicts the optimized space time separable covariance models for September 2023. Exclusively, nested gaussian exponential models were the optimal solution to minimize SSE. Additional figures showing covariance models and SSE are provided in Fig. S1 and S2.

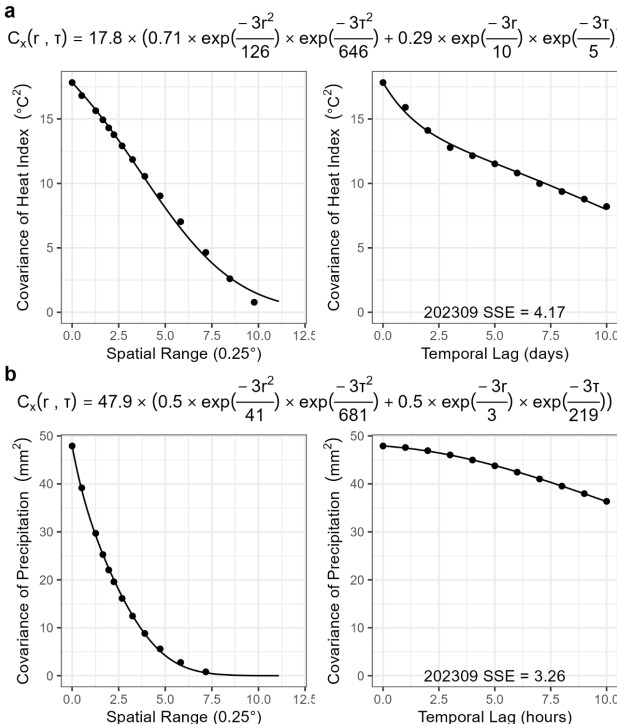

**Figure 4. Optimized covariance models for a) heat index and b) precipitation for September 2023, where the spatial component of each model is described on the left column and the temporal component is shown on the right column. Points represent experimental covariance to which each covariance model was fit. The optimal covariance model is written above each plot. In both cases, a nested gaussian exponential covariance model is the optimal solution for minimizing SSE.**

### 4.2 Space time metric

Space time metrics were calculated using Eq. (4) and are shown in Fig. 5 along with a summarization by month and season. The space time metrics for daily heat index range from 0.42 to 19.39 (median 1.56; summer median 1.23). No months have a space time metric of one, but 148 months have a space time metric less than one. Hourly precipitation space time metrics range from 0.04 to 0.64 (median 0.20). The ratio described by Eq. (2) was not applied to quantify space time metrics because the nested covariance models always outperformed the single structured models when minimizing SSE. Furthermore, Eq. (3) was not applied because the heat index space time metrics range from 0 to 60 following Eq. (3) but range from 0 to 6 (with a single outlier)




following Eq. 4. The results of Eq. (3) reported a tenfold greater spread across the observational dataset; therefore, they were a less robust metric than the results of Eq. 4 (Fig. S3). Further discussion of the SSE, numerators, and denominators that are used to compute the monthly space time metric for each variable is included in the Supplement.

Both an annual and seasonal pattern are observed when aggregating the monthly time series of space time metrics by the month of the year and by the season for both heat index and precipitation (Fig. 5). The space time metric for heat index is relatively low in the winter and summer months whereas the space time metric for precipitation is relatively low in the summer and fall months. There is less overall variability in the precipitation space time metric relative to heat index. The space time metrics for heat index follow a biannual cycle which is higher in the spring and autumn while the precipitation space time metrics indicate an annual cycle oscillating between a summer low and a winter high. However, one cannot draw conclusions about the persistence of a variable in space or time because the space time metric is a ratio, such that changes may be due to changes in the numerator, denominator, or both values.

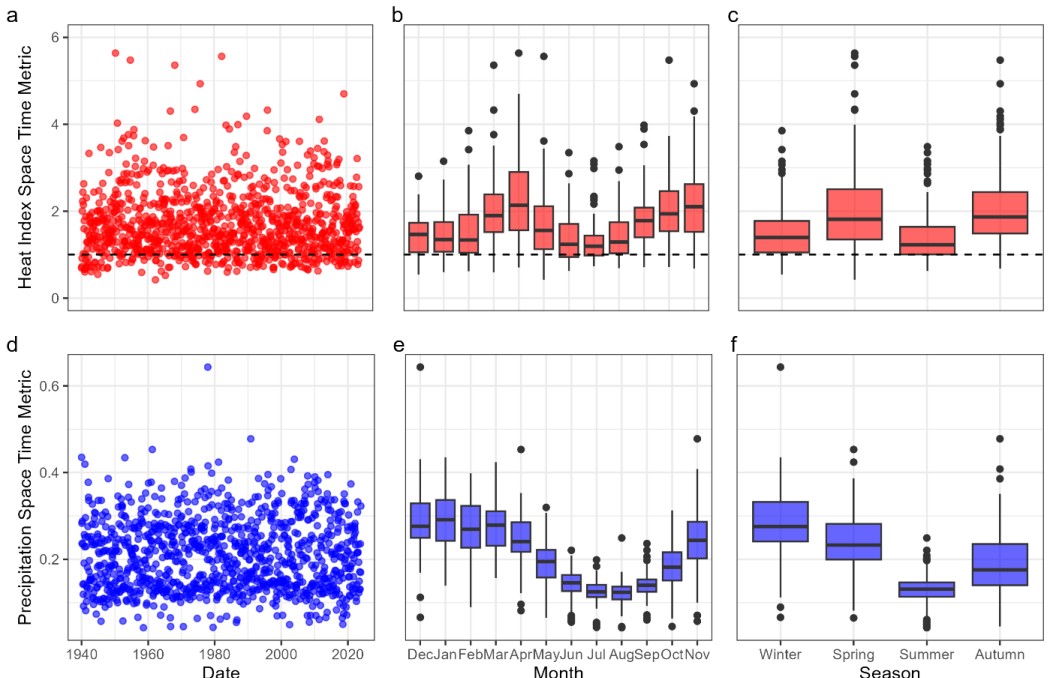

**Figure 5. Space time metrics derived from space time separable covariance models for heat index (red) and precipitation (blue). The horizontal dashed line illustrates a space time metric of one. Metrics are summarized by month (a, d), month of the year (b, e), and by season (c, f). There is a biannual pattern in heat index space time metrics and an annual pattern in precipitation space time metrics. An outlier, 19.39, exists in the heat index dataset, but was removed from plotting to better illustrate seasonality.**

### 4.3 Hypothesis testing

The statistical test for the null hypothesis depends on whether the space time metric meets the assumptions required for parametric testing (i.e., normality and independence). Each dataset is checked for normality using a Shapiro-Wilk test. Since the p-value (heat

index: $p < 2.2e\text{-}16$; precipitation: $p < 7.4e\text{-}13$) is less than the α level of 0.05, we reject the null hypothesis of normality, indicating that the sample does not follow a normal distribution. Therefore, the nonparametric Wilcoxon signed-rank test is more appropriate for hypothesis testing. This test assesses whether the median difference between a sample and the test value is zero (in our case, we are testing whether the space time metric equals one). The Wilcoxon signed-rank test indicated that the median of the space time metrics is statistically significantly different from one ($p < 2.2e\text{-}16$ for both datasets) at a significance level of α = 0.05. The

results of these statistical tests reject the null hypothesis that spatial and temporal distances are equivalent in ERA5-derived daily heat index and hourly precipitation. We therefore find that the native ERA5 resolution is not a space time cube for either daily heat index or hourly precipitation.

### 4.4 Clustering

#### 4.4.1    Space time cube

We develop a space time cube for each dataset using the respective median space time metric. The median space time metric of heat index (1.56) converts a dataset resolved at 0.25 degree per day into a space time cube resolved at 0.39 degree per day. Because extreme heat primarily occurs in the summer, we also apply the summer median space time metric (1.23) to produce a space time cube resolved at 0.31 degree per day. Similarly, the record median space time metric of precipitation (0.20) would necessitate resampling the dataset resolved to 0.25 degree per hour into a space time cube resolved to 0.05 degree per hour. Other space time

metrics (e.g., seasonal medians) were not considered since precipitation events do not exhibit the same interpretable seasonality as heat events. In practice, the heat index dataset is resampled to these coarser resolutions, but the precipitation dataset is not able to be interpolated to a higher resolution. We advise that in cases when the space time metric is below one, a dataset with a higher spatial resolution be acquired to satisfy the assumptions of clustering.

#### 4.4.2    Clustering

To examine the influence of the data resolution on the weather events generated using clustering, we modelled clusters using three different resolutions. The first, 0.25 degree per day (hour), represents the native resolution of ERA5-derived heat index (precipitation). The remaining resolutions, 0.39 degree (0.31 degree) per day, represent the covariance-informed resolution that defines a space time cube across the record (summer season). In each case, observations were compared against thresholds to identify extreme points suitable for clustering into weather events (Fig. 3).



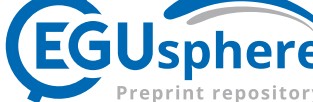

Clustering inputs and outputs are described in Table 1. Coarsening the resolution from 0.25 degree to 0.31 degree (0.39 degree) per day increases the number of heatwaves from 3,261 to 4,384 (3,448). The number of extreme points decreases from 651,183 to 550,107 and 361,679 as the resolution coarsens from an 80 x 63 grid (0.25 degree) to 65 x 51 (0.31 degree) and 51 x 40 (0.39 degree). The neighbour radius similarly decreases from 2.23 (0.25 degree) to 1.41 (0.31 degree and 0.39 degree), indicating that the clustering algorithm is searching a shorter distance to detect neighbouring points. The number of noise points also decreases

from 6,248 (0.25 degree) to 3,343 (0.31 degree) and 3,448 (0.39 degree). In addition to detecting more clusters, spatial coarsening results in clusters with larger extents, but shorter durations than the native resolution of ERA5. The effects of space time metric on the subsequent count, duration, and extent of clusters are illustrated in Fig. S4 and S5.

In theory, the covariance analysis would suggest interpolating the ERA5-derived precipitation dataset to create a space time cube for clustering. However, in practice, interpolation to a higher spatial resolution within an existing dataset will produce an estimation

based on neighbouring observations at the grid level and not enable extrapolation to extreme observations that occurred at the sub-grid level but were averaged across the grid resolution. Therefore, precipitation clustering and comparison with the NOAA Storm Events Database is exclusively conducted at the native resolution of ERA5. The precipitation dataset produces a similar count of clusters (n = 4,780) as for heat index, despite comprising twice as many extreme observations (n = 1,198,384) as any of the heat index datasets. The supplement provides additional analysis of the cluster datasets. We plot the heat index clusters as a function of

time, duration, and exposed area (Fig. S4 & S5).

Table 1. Clustering inputs and outputs. Inputs include the resolution, minimum points, and neighbourhood radius, while outputs include the count of extreme points for clustering, the count of noise points omitted from clustering, and the count of clusters identified. Note that 0.25 degree represents the native ERA5 resolution and 0.31 degree, 0.39 degree, and 0.05 degree are the covariance-informed resolutions for heat index (summer, record) and precipitation (record). Precipitation clustering is not possible for the covariance-

445 informed resolution as a higher resolution dataset is necessary to satisfy the assumptions of spatiotemporal clustering.

| Variable | Description | Resolution | μ | ε | # points | # noise | # cluster | Median Duration (days) | Median Extent (km²) |
|---|---|---|---|---|---|---|---|---|---|
| **Heat Index** | ERA5 resolution | 0.25 degree / day | 4 | 2.23 | 651,183 | 6,248 | 3,261 | 2 | 17,349 |
| | Summer Median | 0.31 degree / day | 4 | 1.41 | 550,107 | 3,343 | 4,384 | 1 | 23,545 |
| | Record Median | 0.39 degree / day | 4 | 1.41 | 361,679 | 3,439 | 3,448 | 1 | 31,600 |



| | | | | | | | | | | |
|---|---|---|---|---|---|---|---|---|---|---|
| **Precipitation** | Record Median | 0.05 degree / hour | - | - | - | - | - | | - | - |
| | ERA5 resolution | 0.25 degree / hour | 4 | 3 | 1,198,384 | 804 | 4,780 | | 2 | 8,674 |

## 4.5 Validation

Cluster and NOAA datasets are reported daily by county from 2019 to 2023 (Fig. 6). Clusters detected using all three resolutions were predicted in most counties across the study region except for the mountainous region of southern Appalachia (Fig 6, top row).

NOAA excess heat episodes were most often reported across Texas, Louisiana, Arkansas, and Oklahoma (Fig. 6, middle row). Fewer episodes were reported in the Midwest and were rarely or never reported elsewhere during this period. Clusters detected using the covariance-informed space time cubes were predicted with a similar spatial pattern but affected a larger area of the Appalachian region as a function of coarser spatial resolution (Fig. 6). Clusters were detected along the domain's western boundary when using the native ERA5 resolution, but not after coarsening the resolution of the data using the covariance-informed method.



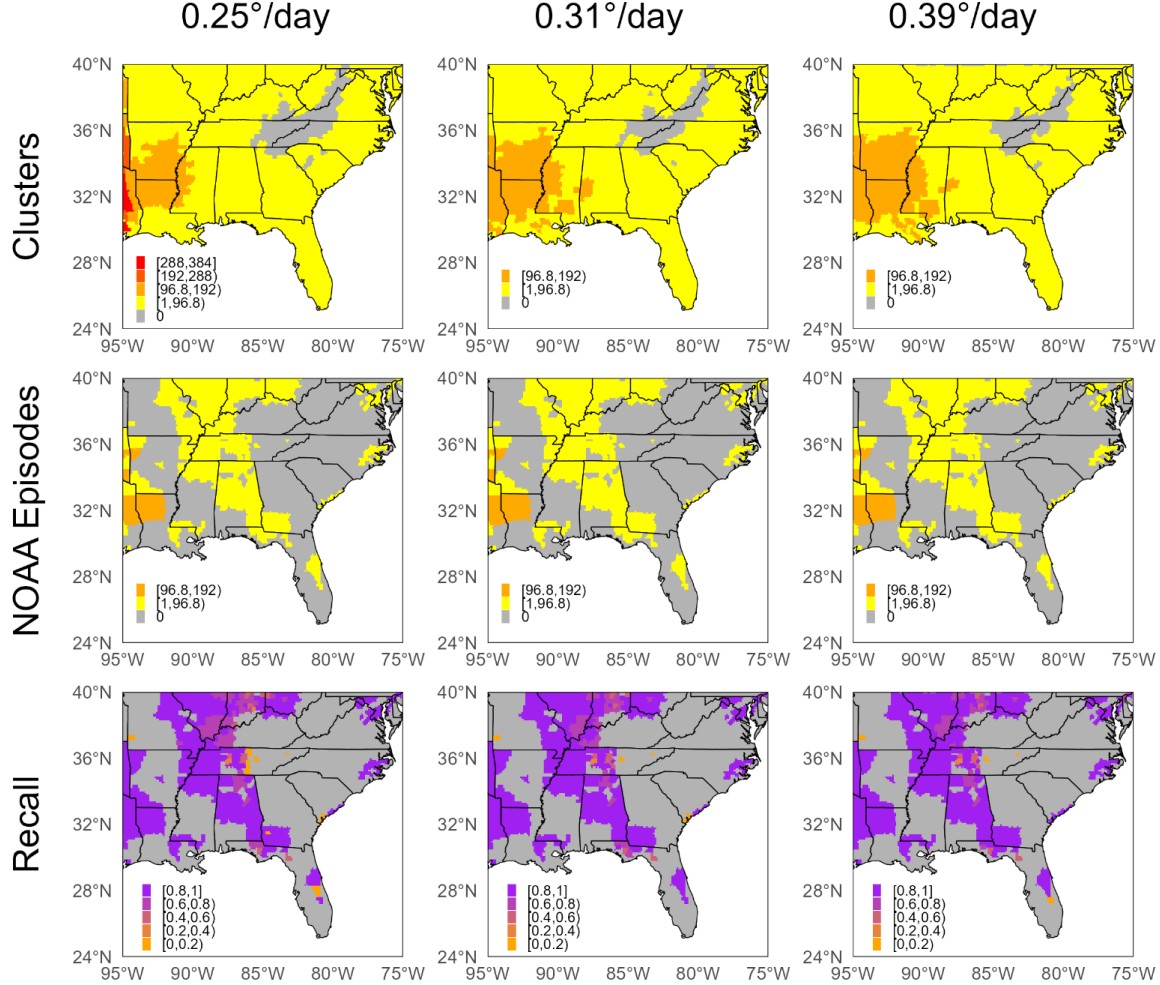

**Figure 6. Comparison of heat index clusters and NOAA excess heat episodes (2019-2023). The top row illustrates the count of days that each county observed a heat index cluster. The middle row describes the count of days that each county experienced a NOAA excess heat episode. The bottom row represents the county-specific recall of NOAA episodes by clusters. The columns represent the resolution of the dataset for clustering, which increases from the native resolution of ERA5 (0.25 degree per day), to the resolution determined with the space time metric summer (0.31 degree per day) and record median (0.39 degree per day). Overall recall increases from 0.92 for the ERA5 dataset to 0.94 for both space time cubes.**

Recall of NOAA excess heat episodes is calculated from a confusion matrix (Table S4, S5, and S6) using both the native ERA5 resolution and the covariance-informed space time cubes (Fig. 6, row 3; Table 2). Recall across all county-days improves from 0.92 to 0.94 when the native ERA5 resolution (0.25 degree per day) is resampled into space time cubes at both the summer and record median resolutions (0.31 degree and 0.39 degree per day). Recall is higher for weather events reported in more recent years (Table 2). This finding is further supported when recall is aggregated across the domain per day to calculate a time series of model performance across the SEUS (Fig. S6).



**Table 2. Recall of NOAA heat episodes with clustering. The ability for a model to perfectly reconstruct an observed dataset would result in a recall of 1. Recall is calculated for clustering datasets using the native ERA5 resolution (0.25 degree per day), the resolution determined with the space time metric summer (0.31 degree per day) and record median (0.39 degree per day). Recall is also calculated for different periods of the NOAA dataset. Both space time cubes report the greatest recall (0.94) for 2019-2023.**

| Time Period | Recall | | |
|---|---|---|---|
| | 0.25 degree per day | 0.31 degree per day | 0.39 degree per day |
| **2000-2023** | 0.66 | 0.69 | 0.70 |
| **2010-2023** | 0.77 | 0.80 | 0.81 |
| **2019-2023** | 0.92 | 0.94 | 0.94 |

The validation of precipitation clusters at the native ERA5 resolution has a recall ranging from 0.03 to 0.36 (Fig. S7, Table S3). This relatively poor validation may be due in part to the underestimation of extreme precipitation by ERA5 as well as concerns of the NOAA Storm Events Database involving possible delays in reporting (Dougherty & Rasmussen, 2019; Hassler & Lauer, 2021).

## 5. Discussion

### 5.1 Covariance modelling

This study offers the first empirical approach to proportionally transform spatial distances into equivalent temporal distances prior to conducting spatiotemporal clustering. We show that space time separable covariance modelling is a useful tool for understanding and reducing bias in the spatiotemporal clustering framework. Space time metrics were calculated for maximum daily heat index and hourly accumulated 24-hour precipitation. We hypothesized that a space time metric of one indicates a dataset is already formatted as a space time cube, and when the space time metric is not equal to one, that the spatial resolution of the dataset can be scaled by this factor to form a space time cube. Neither dataset displayed space time metric equal to one, suggesting that neither dataset can be considered a space time cube outright. We suggest that while it is possible then to coarsen ERA5-derived heat index, since the space time metric for ERA5-derived precipitation is less than one, it is not practical to interpolate and achieve a greater data resolution. In this sense, a higher resolution dataset would be preferred for clustering precipitation extremes to prevent biased clustering in either dimension. The space time metric therefore quantifies the degree of bias in the resolution of a dataset for the purpose of spatiotemporal clustering. While clustering the precipitation dataset is still possible, clustering should proceed with the highest resolution data available even if the resulting clusters may have an incorrect count, characteristics, and IDF curves. At a minimum, the quantification of a space time metric provides insight into whether clustering results are biased in space or time.

We also demonstrate that a covariance-informed approach for defining the space time metric enables physical interpretations of the statistical trends in fluctuations observed over space and time. We find that nested covariance models reduce SSE to a greater

degree than single covariance models, which implies that both heat index and precipitation processes display nested fluctuations. We interpret this to mean that each variable experiences physical fluctuations over at least two temporal scales (short and long duration processes) and two spatial scales (local and regional-to-global processes) per month in the SEUS. The space time metrics for heat index and precipitation both display seasonal variability. Heat index values describe a biannual pattern while precipitation values follow an annual trend. This is unsurprising since there is a seasonal cycle to precipitation availability as shown by the annual water year and a biannual cycle to temperature oscillations between the summer and winter. Future work could disentangle the relative influence of short lived and local weather patterns, seasonal and hemispheric variability, and decadal to global climate oscillations that may influence these distinct covariance structures.

Understanding the physical processes controlling space time metrics may prove critical for interpretation of the scale of detected weather events in other studies. Physical processes may explain why the space time metrics tend to be lower for daily heat index during the summer and winter and for hourly precipitation during the summer and autumn (Fig. 5). Since the space time metric is the ratio of spatial to temporal persistence, we interpret a lower space time metric to describe a lesser persistence in space relative to time, while a higher metric implies the opposite. This suggests that heat index variability is low during the summer and winter seasons and would support consistently warmer, more moist summer and colder, drier winter conditions. Conversely, heat index variability is greater in the transitional seasons. We interpret the space time metric variability to be explained by the seasonal availability of heat and moisture, primarily driven by the activity of synoptic systems in the coastal mid-latitudes governing both the amplitude and duration of humid heat anomalies. We predict seasonal differences in the space time metric of heat index scale with synoptic activity. That is, we would expect to see the greatest absolute differences in the transitional months where the meridional temperature gradient and associated baroclinicity strengthen as the atmosphere shifts between cold and warm season configurations. More frequent and intense extratropical cyclones and frontal passages would likely occur under enhanced baroclinic instability, injecting alternating pulses of warm, moist air with colder, drier air masses (Lembo et al., 2017). Furthermore, daily temperature ranges peak in the mid-latitudes during these transitional seasons, reflecting large, short-lived swings in temperature and moisture fields (Leathers et al., 1998). In contrast, midsummer is dominated by the North Atlantic subtropical high and persistent maritime-tropical (mT) air masses, while midwinter is governed by continental-polar (cP) ridging. These quasi-stationary regimes lead to smaller deviations from the seasonal mean that are maintained over longer durations. Furthermore, the moist bulk stability (i.e., the potential temperature gradient from surface to tropopause) peaks in both summer and winter across the mid-latitudes, reflecting a stable atmosphere that suppresses rapid fluctuations in humidity and temperature (Frierson & Davis, 2011).

We interpret relatively low precipitation variability during the summer and autumn to describe more frequently wetter conditions often driven by convective processes than in the winter and spring when episodic precipitation is primarily driven by frontal

systems. Relatively consistent summertime precipitation may be associated with mesoscale convective systems whereby uniformly warm, moist air, coupled with surface heating leads to frequent convective events (J. Li et al., 2021; Schumacher & Rasmussen, 2020). Conversely, more variable wintertime precipitation may be associated with cyclonic activity and frontal passages that produce sporadic and often extreme precipitation (Bishop et al., 2019; Konrad, 1997). We predict seasonal differences in the space

time metric of precipitation correlate with the scale of precipitation organization (Rickenbach et al., 2015). We would expect to see the greatest absolute differences between seasonal space time metrics in regions that observe a large annual variability in storm frequency and the lowest differences in regions that observe either consistently wet or dry conditions.

Disentangling trends in the numerator and denominator is necessary for interpreting the subsequent space time metric. The space time metric will not vary if the persistence of a variable in space and time changes at similar rates. A consistent space time metric

may either mean that the physics of the system are not changing over time or that the scale of the fluctuation in space and time are both changing at an equivalent rate. In any case, understanding how the numerator (spatial fluctuation) and the denominator (temporal fluctuation) change over time are necessary for interpreting the behaviour of a space time metric (Fig. S2). Furthermore, we quantified a single space time metric across the domain per month for each variable. However, we expect that there is variance in the physical processes controlling the scale of atmospheric fluctuations across climatologically heterogeneous regions when

evaluating such a diverse domain as the SEUS. Future analysis may benefit from quantifying space time metrics across climatologically homogenous regions to ensure that fluctuations quantified using geostatistics are indeed representative of the physical processes controlling a given weather event (e.g., regions exclusively exposed to convective precipitation may require a different space time metric than regions exclusively exposed to stratiform precipitation).

## 5.2 Spatiotemporal clustering

The smallest possible cluster contains four points, which range in three dimensions from a spatiotemporal extent of one timestep containing four neighbouring locations (e.g., points in space) to four timesteps at a single location. We detect 3,261 clusters of extreme heat index observations and 4,780 clusters of extreme precipitation observations. We expect that, compared to heat index, we identify more precipitation clusters relative to the number of extreme points because of the artificial autocovariance incorporated due to the rolling window approach. In this sense, extreme precipitation is more likely to occur in sequence than daily

extreme heat index due to the dependent nature associated with the rolling sum of 24-hour precipitation, as opposed to daily maximum heat index which is calculated independently of previous timesteps.

The count of clusters does not linearly increase as the resolution is coarsened. While more heat index clusters are detected in both covariance-informed space time cubes, the count of clusters is a function of the resolution, the count of extreme points, and the

neighbour search radius. The summer and record median space time metric for heat index (1.23 and 1.56) coarsens the ERA5

resolution from a 80 by 63 grid to a 65 by 51 and a 51 by 40 grid, respectively. This coarsening results in fewer extreme observations

that are closer to other extreme observations ($\varepsilon = 1.41$) than in the original resolution ($\varepsilon = 2.23$), resulting in more heatwaves

despite fewer extreme observations. We also observe that in each year a single cluster of extreme heat index values can persist for

most of the summer season, suggesting that the thresholds for extreme observations of heat index may currently be set too low for

detecting acute, rather than the currently observed chronic, heat hazards. Future analysis could revise this threshold using either a

555 percentile of historic observations or a higher threshold informed by impacts, such as the excessive heat warning rather than the

heat advisory applied in this study.

### 5.3 Validation

Reported NOAA excess heat episodes are not evenly distributed across the SEUS. Some Weather Forecast Offices (WFOs) never

report events, while some infrequently or inconsistently report, but reporting frequency is increasing overall since records began.

Out of 43 NOAA WFOs in the study area, most NOAA excess heat episodes were reported by four WFOs from 2019 to 2023:

Shreveport, LA (SHV), Tulsa, OK (TSA), Memphis, TN (MEG), and Paducah, KY (PAH) (Fig. 6). This suggests bias in the

reporting to NOAA Storm Events Database. WFO's are responsible for reporting storms to this database, and over time, individual

WFO's may change their criteria for defining storms or begin reporting with greater frequency. As a result, the quality of episode

reporting has improved since records became established some thirty years ago. Consequently, there is an increased recall when

comparing the clusters against weather events reported in more recent years than over the entire record.

Most hazardous days predicted with spatiotemporal clustering are found across the states of TX, OK, LA, AR, MS, and AL (Fig.

2). Few clusters are reported across the FL peninsula, even though it is exposed to chronic heat conditions (Diem et al., 2017). We

infer that these chronic heat conditions, while still harmful to human health, are below the detection limit of our chosen thresholds

which are based on regionally defined criteria. Analysis seeking to cluster chronic heat conditions may select a more conservative

threshold that accounts for human health effects or regional tolerance. No clusters are predicted across the Appalachian region.

This is perhaps because all seven WFOs in NC define hazardous heat conditions using the same empirical threshold of heat index,

irrelevant of the geographic or topographic characteristics. Yet, it is unlikely that residents across the entire state are acclimated to

the same hazardous heat criteria and suggests that future research may further improve upon this assessment by defining more

spatially resolved, or impact-informed, thresholds (Raymond et al., 2020; Xu et al., 2018).

The validation shows that spatiotemporal clustering of heat index at the native ERA5 resolution has a recall of 92%, demonstrating

that impactful heat waves can be detected with traditional clustering methods. However, clustering using a covariance-informed

resolution (space time cube) further improves recall of NOAA excess heat episodes to 94% at a daily, county-wide resolution across the SEUS from 2019 to 2023. While the improvement is relatively small, this finding demonstrates that, by first creating a space time cube using covariance modelling, spatiotemporal clustering is even more reliable for detecting areas exposed to

580 impactful heat waves. Notably, the low recall of precipitation-driven hazards using the native resolution dataset (Fig. S7, Table S3), alongside the knowledge that a dataset with at least a five-fold higher spatial resolution is needed for unbiased spatiotemporal clustering (Fig. 5), provides further evidence that ERA5 is not sufficient for evaluating severe storms in this region.

Importantly, our results suggest that previous studies that have clustered ERA5-derived storms and/or heat waves at regional to global scales using the data at its native resolution (Liu et al., 2023; Liu & Zhou, 2023a; Luo et al., 2022; Tilloy et al., 2022) may

result in finding different weather event sets or different weather event characteristics than if space time separable covariance modelling were to be applied to the datasets prior to clustering. For example, in the SEUS, a larger space time metric generates more, shorter lasting, larger heatwaves than would be detected using the native ERA5 resolution dataset. The heatwaves produced with a larger space time metric more accurately match historically observed heatwaves. We also find that in the SEUS, severe storms clustered using ERA5-derived precipitation and a space time metric of one do not identify historically observed storms or

flooded areas. We would expect recall to improve if a smaller space time metric were instead applied. Furthermore, the space time metric for precipitation implies that previous application of spatiotemporal clustering to ERA5-dervied precipitation is likely underestimating the true number of storm events or missing their physical characteristics altogether, especially smaller or short-lived storms such as thunderstorm complexes and mesoscale convective systems that are not as well resolved at the resolution of ERA5 as are stratiform and synoptic scale systems. This suggests that datasets where the resolution was not checked prior to

clustering should be used with caution as they may be biased in either the spatial or temporal dimension.

### 6. Conclusions

This study offers a quantitative test to test the assumptions of spatiotemporal clustering with respect to a dataset of interest. We demonstrate that space time separable covariance modelling can be used to empirically derive a space time metric and test whether a dataset satisfies the definition of normalization for spatiotemporal clustering. If the dataset does not satisfy this definition, space

time separable covariance modelling offers an empirical solution for determining the appropriate resampling factor to produce unbiased clustering results. We find that neither ERA5-derived heat index nor precipitation currently satisfy the requirement that distance in space is equivalent to distance in time, and that ERA5-derived heat index data must be coarsened from 0.25 degree to 0.31 degree or 0.39 degree per day to prevent preferential clustering and demonstrate that this improves the recall of NOAA excess heat episodes from 0.92 to 0.94. Furthermore, we find that ERA5-derived precipitation data is too coarse, and that an alternative

dataset with a resolution of 0.05 degree per hour is necessary to satisfy the normalization required for unbiased spatiotemporal clustering.

We present the first application of spatiotemporal clustering that satisfies this definition using space time separable covariance modelling. This method resolves two methodological gaps. First, covariance-informed spatiotemporal clustering quantifies the necessary variable-specific spatial resolution for a predefined temporal resolution, thereby enabling a researcher to determine early

in a project whether a dataset satisfies this important assumption of spatiotemporal clustering. Second, covariance-informed spatiotemporal clustering improves the researcher's ability to detect weather events over traditional clustering approaches. We find that the spatial resolution differs significantly by variable, with hourly precipitation requiring a resolution eight-fold higher than for daily heat index. We expect that the covariance-informed spatial resolution will vary with the variable of interest, temporal resolution of the dataset, period of evaluation, and domain of study, as the persistence of a given variable likely changes under

each of these criteria.

We argue that the detection of weather events from reanalysis data may improve if researchers incorporate covariance-informed spatiotemporal clustering into their research frameworks so that they may critically assess whether they are satisfying the conditions of unbiased spatiotemporal clustering prior to implementation. We demonstrate that this approach not only can be applied to various geophysical observations when clustering observations, but crucially that this approach yields superior performance

metrics. Ultimately, the advantage of this approach is that covariance-informed spatiotemporal clustering may be applied to any dataset to quantify, and potentially reduce, bias in event detection. Improved event datasets will lead to better understanding of extreme weather intensity, duration, extent, and frequency, but also in hazard modelling and impact analyses.

**Code availability**

The code used in this study is available at https://github.com/hquintal16/phd1_cluster_southeast under an MIT license.

**Data availability**

This analysis was conducted using R version 4.4.3, MATLAB version R2024a, and Python version 3.9.12. The meteorological datasets analysed are available through the ECMWF Reanalysis v5 (ERA5) repository, https://www.ecmwf.int/en/forecasts/dataset/ecmwf-reanalysis-v5. Heat index thresholds are available in the NWS Directives repository for the Central and Eastern regions, https://www.weather.gov/directives/010 and in an online repository for the Southern

region, https://noaa.maps.arcgis.com/apps/MapJournal/index.html?appid=964c64e07e2d43629d6d7e17facd85fc. Precipitation threshold datasets are available through the NOAA NWS Hydrometeorological Design Studios Centre (HDSC) Precipitation

Frequency Data Server (PFDS) repository, https://hdsc.nws.noaa.gov/pfds/. NOAA NWS WFO CWA shapefiles are available

through the NWS online GIS portal, https://www.weather.gov/gis/CWABounds. NOAA storm events are available in the NOAA

NCEI Storm Events Database repository, https://www.ncdc.noaa.gov/stormevents/. Environmental Protection Agency (EPA)

Ecoregion shapefiles are available through the National Aquatic Resources Survey, https://www.epa.gov/national-aquatic-

resource-surveys/ecoregions-used-national-aquatic-resource-surveys. BMElib software is available in the BMElib repository,

https://mserre.sph.unc.edu/BMElib_web/.

**Author contribution**

HQ: Conceptualization; Data curation; Formal analysis; Investigation; Methodology; Software; Validation; Visualization; Writing

– original draft preparation; Writing – review & editing

AS: Conceptualization; Funding acquisition; Resources; Methodology; Supervision; Validation; Visualization; Writing – original

draft preparation; Writing – review & editing

MS: Conceptualization; Resources; Methodology; Software; Supervision; Writing – review & editing

WJ: Conceptualization; Methodology; Writing – review & editing

MCR: Resources; Writing – review & editing

**Competing interests**

The authors declare that they have no conflict of interest.

**Acknowledgements**

We thank NOAA NWS Public Weather Services Program Manager Kimberly McMahon for providing access to a repository

required for the analysis. We thank the Myriad-EU project at Vrije Universiteit Amsterdam for hosting H. Quintal as a visiting

researcher during the summer of 2024. We thank members of the Myriad-EU project for constructive discussions, especially

Davide Ferrario, Kelley de Polt, Judith Claassen, and Sophie Buijs.

**Financial support**

This research was supported by the National Oceanic and Atmospheric Administration (NOAA) through the NOAA Climate

Adaptation Partnerships (CAP) program: Innovating a Community-based Resilience Model on Climate and Health Equity in the

Carolinas (2021–2026) (NA21OAR4310312). MR and WJ also received funding from the MYRIAD-EU project, which received





funding from the European Union's Horizon 2020 research and innovation programme under grant agreement No. 101003267.

MCR also received support from the Netherlands Organisation for Scientific Research (NOW) (VENI; grant no. VI. Veni.222.169).

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
