# Peer review of "Covariance-informed spatiotemporal clustering improves the detection of hazardous weather events"

_EGUsphere, 2025_

## Author Comment (AC1)

**Referee #1:**

General comments: The approach the authors offer exploring debiasing spatial and temporal covariance as a preprocessor for cluster analysis is an interesting idea that has not been explored in atmospheric science. The idea certainly has merit, and I found no major issues with the methodology as presented. However, the application of the methods in sections 3 and onward had some concerns. First, the authors quantify a "heat wave", which is typically reserved for a prolonged period of excessive heat, as any extreme heat measure as quantified by heat index. This created a few issues, as I discuss below. Second, and in my opinion the most glaring issue, the authors did not explore the importance of false positives in a database they already noted was rare-event. Without quantifying false positive rates in some way, the authors are presenting a model that may actually have little to no skill. I discuss this further below. Finally, the authors, despite providing so much statistical detail, did not provide any measure of significance in their recall performance statistics in their demonstration section.

Most of the issues can be addressed through some additional thought or analyses in the case-study part of this work. Without a stronger case in that section, it is hard to see the inherent value in adding complexity to the problem with the debiasing preprocessor. If however results show significant differences, the authors may have identified a method that could be helpful for cluster analysis studies going forward.

Based on these comments, I recommend major revisions.

We would like to thank the reviewer for their thoughtful feedback. We plan to revise the manuscript to address both reviewers' comments by expanding the discussion of selected variables and model parameter sets, elevating the results and discussion of precipitation clusters to the main text from the supplement, adding discussion of the potential for space time metrics to quantify regional climatological processes, calculating false positive rates to compliment the model skill already described with recall, and quantifying whether the improvement observed in heat wave recall is statistically significant using an ANOVA test.

To address the reviewer's concern about our use of the term 'heat wave', we plan to refer to these as 'extreme heat events' in a revised version. In addition, we will further explore the issue of false positives in the database by calculating daily false positive rate across the three different spatial resolutions and will include an additional supplementary table that reports the performance statistics by Weather Forecast Office (WFO). In doing so, we will be able to quantify the false positive rate in regions where we have high confidence that heat events are reported. We will use an ANOVA test of daily recall by WFO to quantify

whether the observed improvement is statistically significant. We will explain how the chosen variable (e.g., 1- vs 24-hour precipitation) affects the calculation of space time metric and explain how the chosen model parameter set (e.g.,  $\mu$  and  $\epsilon$ ) influences the clustering of extreme observances. We will elevate the precipitation results and discussion to the main text for comparison of space time metrics across variables (e.g., heat index and precipitation) to address concerns about the lack of discussion of this variable. We will also include a subsection of the discussion dedicated to describing the value that space time metrics offer by potentially quantifying physical processes at the regional scale.

We have addressed the reviewer's comments in a point-by-point style below. Our responses can be found in red.

**Major comments:**

The use of the heat index as a measure of heat waves can introduce challenges if you use the heat index data outside of its intended ranges, which is likely your source of unphysical values. Did you consider using wet bulb globe temperature, which is used by many SEUS public entities to measure human health risk from heat? It may be a better measure than heat index for this type of application, though it is challenging to compute. The data in ERA5 should be sufficient to obtain this measure and remove the NA issue.

In this study, we chose to use heat index to threshold the data and initiate extreme heat clusters because NOAA Weather Forecast Offices produce heat advisories and extreme heat warnings for heat index. While we recognize WBGT is perhaps a better measure of human health impacts from heat, NOAA Weather Forecast Offices do not issue advisories or warnings based on WBGT. Extensions of this work with other heat impact measurements and thresholds could present interesting comparative analysis.

Why are you extrapolating extreme heat index and precipitation values over the ocean, where such things are not defined? Does your method require spatially continuous data, or could you use a land-sea mask and focus on the actual CWAs of the region. It seems continuity is important, so this may be a limitation of your approach.

We extrapolate thresholds over the ocean because the clustering method requires spatially continuous data. This is especially important where few ERA5 grid cells overlay land, including edge cases such as coastal regions and peninsulas where hazardous weather events are common but would be overwhelmingly masked from the resulting cluster dataset(s). We expect that without extending thresholds seaward in these regions, we would be unable to define threshold exceedances and therefore be prevented from (or at least biased low when) clustering exceedances.

The choice to not penalize false positives (lines 355-358) is problematic. This speaks to a major issue with the Weather Service, the false alarm problem. It is critical, if this method is to be evaluated against existing methods in a fair manner, that you at least quantify the rate of false alarms (false positives) relative to your projections. That is, if your model always identified a heat wave, it would have a very high recall, but its false positives would be enormous, and the model would have no skill. If you choose to utilize confusion matrices to evaluate performance of your clustering, this is a major limitation of your work.

Thank you for pointing this out. We agree with both reviewers that both recall and false positive rate should be included to discern model skill in a revised version of the manuscript. We originally chose to only report recall because we discovered during the initial analysis that NOAA storm event database (validation source) quality is heterogeneous in space (WFOs report differently than their neighbors) and in time (WFOS report differently under different leadership). In response to reviewer 2, we will calculate recall and false positive rate per WFO and add this within a supplementary table to enable a comparable model performance to adjust to the observed heterogeneity of the validation source. We will also add more discussion within the manuscript to address the fact that some WFOs do not report. In addition, we will include two figures (below) to the supplementary materials that visualize the count of NOAA storm events by Weather Forecast Office. We utilize the NOAA storm events database as a validation source because it is considered the best available database of weather events for CONUS, yet the figures attached below indicate that the quality of reporting is actually quite varied spatially.

Count of Heat and Excessive Heat events reported to the NOAA Storm Events Database between 1950 and 2023. Regional outlines represent NOAA Weather Forecast Offices. NOAA Storm Events may be reported by County, State or by Weather Forecast Office. There is spatial heterogeneity in which Weather Forecast Offices report heat related hazards.

Count of Flood, Flash Flood, Heavy Rain, Hurricane, Tropical Depression, Tropical Storm, and Typhoon (Hurricane) events reported to the NOAA Storm Events Database between

1950 and 2023. Regional outlines represent NOAA Weather Forecast Offices. NOAA Storm Events may be reported by County, State or by Weather Forecast Office. There is spatial heterogeneity in which counties report heat related hazards, which tend to be the location of large urban metro areas.

With all the statistical analysis provided in the text, it was surprising that Table 2, which is a key result in this paper, did not contain some measure of significance in the differences of the results of recall. The results are so similar I wonder if there is statistically significant benefit to this complex approach over traditional approaches. This would strengthen your results if they were significant.

We agree with the reviewer that the improved results when resampling heat index do not appear to be significantly different from the original heat index. We will apply ANOVA testing to compare the daily recall and false positive rate between the two resolutions to quantify whether there is significance in this improvement. However, we feel it is important to underscore that we expect that the scale of the calculated space time metric will inform the expected improvement of the resampled clustering. For example, the heat index space time metric is greater than 1 by a factor of 1.5 to 2, while the precipitation space time metric is less than 1 by a factor of 3 to 10. We would therefore expect that the amount of improvement would be greater for precipitation than for heat index. Since only the heat index space time metric enables resampling, we do not expect a significant difference in results, but conversely, we would expect the precipitation results would be significantly different if the resolution of the precipitation data were higher.

There may be value in exploring the 4-6% of "heat waves" that were missed by your methodology, since missing a heat wave seems like something that should not really happen in reality. Why were those 6% missed, especially when you clearly have a major false positive issue already per figure 6.

We agree with the reviewer that 4-6% of missed heat events are a valid and interesting concern. We agree that these 4-6% of missed heat events should not really happen in reality. We expect to find that the discrepancy in missing events is due to the criteria set by a select few WFOs when issuing heat advisories – specifically the advisories posted by WFOs that consider the temperature vs heat index variable or varied intensity-duration-frequency definitions (including seasonal deviations) for defining exceedances.

**Minor comments:**

A heat wave is defined as an "unusually warm and unusually humid weather, typically lasting two or more days" according to NOAA. However, your approach only looks at daily

heat index values. How can you relate these results to an actual heat wave? **Or is a better approach to state you are measuring "extreme heat"?**

Thank you for pointing this out. To address your concern, we plan to refer to the clusters as extreme heat events instead of heat waves throughout the manuscript.

The use of both  $\alpha$  and a in denoting important variables for deriving your metric is a bit confusing. The reader has to look closely to tell which are  $\alpha$  and which are a. If possible, consider using a different variable to represent one or the other to ease differentiating the two when discussing the equations.

Thank you for this comment. We acknowledge that using these two variables is similar. However, to be consistent with previous literature, we are using  $\alpha$  and a (see Christakos, G., Hristopulos, D., & Bogaert, P. (2000). On the physical geometry concept at the basis of space/time geostatistical hydrology. *Advances in Water Resources*, *23*, 799–810.).

There appears to be some sort of typo on lines 241-246 in the text in terms of formatting.

We agree that this is a formatting issue and will fix it in the revised version.

It is a little strange to delineate hurricanes and tropical storms on line 250. It would be better to just say "tropical cyclones".

We used the same storm types that the NCEI NOAA Storm Events Database uses in their reporting, however, we agree with the reviewer's comment. In our revision, we will group all tropical depressions, storms, and cyclones into a singular tropical cyclone category.

I may have missed it, but is the variable  $\epsilon$  defined on line 334? If not, it should be explicitly defined in the text.

The variable  $\epsilon$  is defined within the caption of Figure 1 but is not explicitly defined in the text. We will update the methods section in the revised manuscript to include a definition for  $\epsilon$ .

**Referee #2:**

**General Comment**

The authors present a rigorous and innovative methodological framework for improving the detection of hazardous weather events using covariance-informed spatiotemporal clustering. The study addresses a critical gap in the literature: the lack of standardized approaches to test whether a dataset's spatiotemporal resolution is appropriate for unbiased clustering. The manuscript is well written, methodologically sound, and contributes meaningfully to the field of extreme weather event detection.

However, I share some concerns with reviewer 1 regarding the application of the method to reanalysis data. While the technical novelty of the covariance-informed space-time metric is clear, the current manuscript lacks information to justify how this approach improves the characterisation of spatiotemporal features of extreme events. Furthermore, the article (at least the main part) does not leverage the huge amount of work done by the authors to create clusters for the period 1940-2023 over the Southern USA.

We would like to that the reviewer for their thoughtful feedback. We plan to revise the manuscript to address both reviewers' comments by expanding the discussion of selected variables and model parameter sets, elevating the results and discussion of precipitation clusters to the main text from the supplement, adding discussion of the potential for space time metrics to quantify regional climatological processes, calculating false positive rates to compliment the model skill already described with recall, and quantifying whether the improvement observed in heat wave recall is statistically significant using an ANOVA test.

We will further explore the issue of false positives in the database by calculating daily false positive rate across the three different spatial resolutions and will include an additional supplementary table that reports the performance statistics by Weather Forecast Office (WFO). In doing so, we will be able to quantify the false positive rate in regions where we have high confidence that heat events are reported. We will use an ANOVA test of daily recall by WFO to quantify whether the observed improvement is statistically significant. We will explain how the chosen variable (e.g., 1- vs 24-hour precipitation) affects the calculation of space time metric and explain how the chosen model parameter set (e.g.,  $\mu$  and  $\epsilon$ ) influences the clustering of extreme observations. We will elevate the precipitation results and discussion to the main text for comparison of space time metrics across variables (e.g., heat index and precipitation) to address concerns about the lack of discussion of this variable. We will also include a subsection of the discussion dedicated to describing the value that space time metrics offer by potentially quantifying physical processes at the regional scale.

In addition, we address the reviewer's comments in a point-by-point manner below.

Thank you for pointing this out. This is a fair critique, and one that requires comparison to a higher resolution precipitation database to demonstrate improvement in model skill, especially for hazardous weather events that are not resolved in a coarse product such as ERA5 or NOAA Events Database. However, improvement is shown for heat index, despite a small change in space time metric. We will elevate the precipitation results to the main text to compare these two variables. As discussed in our response to reviewer 1, we agree with the reviewer that the improved results when resampling heat index do not appear to be significantly different from the original heat index resolution. We will apply ANOVA testing to compare the daily recall and false positive rate between the two resolutions to quantify whether there is significance in this improvement. However, we feel it is important to underscore that we expect that the scale of the calculated space time metric will inform the expected improvement of the resampled clustering. For example, the heat index space time metric is greater than 1 by a factor of 1.5 to 2, while the precipitation space time metric is less than 1 by a factor of 3 to 10. We would therefore expect that the amount of improvement would be greater for precipitation than for heat index. Since only the heat index space time metric enables resampling, we do not expect a significant difference in results, but conversely, we would expect the precipitation results would be significantly different if the resolution of the precipitation data were higher.

I believe the manuscript has strong potential but would benefit from major revisions. Below, I outline specific areas for improvement.

- 1. Overlooked methodological aspects
  - The results of spatiotemporal clustering with DBSCAN are highly sensitive to the parameter set, and to the thresholds used (Tilloy et al., 2022). In section 3.2, you set threshold for extreme events based on NOAA datasets. These thresholds are impact-relevant but they may induce over or under sampling of extremes in the ERA5 data due to major differences between the underlying data in NOAA datasets and ERA5.

    This may explain the poor recall for precipitation extremes. Thank you for this comment. Indeed, this may further explain the poor recall of precipitation extremes, where a lower threshold (or, conversely, a shorter duration) may result in improved validation. However, we elected to use the 1-year return period, 24-hour duration because of its relevance to stormwater infrastructure design in the U.S. We will include additional information in the discussion to address this point. The revised methods will include: 1) the parameter set was determined to conservatively include as

many clusters as possible by setting  $\mu$  = 4 and defining  $\epsilon$  with  $k^{th}$  nearest neighbor analysis, and 2) thresholds were defined to conservatively include as many exceedances with the potential to be clustered as possible by selecting the lower of two impact-relevant heat index thresholds (heat advisory vs excessive heat warning) provided by NOAA WFOs and the lowest of all possible 24-hr precipitation thresholds (1-year vs the 2, 5, 10, 25, 50, 100, 200, 500, or 1000-year precipitation) provided by NOAA Atlas 14.

- The choice of aggregating precipitation with a rolling sum is justifiable from an impact perspective. From a clustering perspective, it may result in an overestimation of temporal covariance, biasing the conclusions regarding the recommended spatial downscaling.
- o This is a fair concern; however, the covariance model is applied directly to the preprocessed precipitation data. Therefore, with the same underlying data, a different space time metric will be calculated for 1, 3, 6, 12, and 24hour precipitation, etc. We will address this point in the discussion, specifically how one defines a variable that will influence the subsequent space time metric. The 1-hour temporal covariance model will contain more hourly variability than the 24-hour model, resulting in a smaller denominator and therefore a larger space time metric. We chose the 24-hour precipitation to compliment the resolution of the validation dataset as well as the design standards for infrastructure projects in the United States. We agree that the 24-hour precipitation will indeed bias the clustering results by potentially producing multiple exceedances at the same location over a period of time. This is in part intentional because isolated exceedances in 1-hour precipitation would possibly result in an undercount of clusters, while we expect 24-hour precipitation exceedances to more reliably cluster. We further coarsen clusters to a daily resolution for validation purposes.
- The lack of precipitation validation is a major limitation. The authors state that a higher-resolution dataset is needed, but they could discuss available downscaling techniques. Furthermore, could an upscaling of temporal resolution have been an option to overcome the issue?

We will elevate the precipitation validation analysis from the supplement to the main text. We will include discussion of available downscaling techniques to resolve this issue, with points including 1) statistical downscaling of ERA5 spatial resolution and 2) upscaling of ERA5 temporal resolution to overcome these limitations. We elected not to upscale the

temporal resolution because this would likely bias low the count of clustered events in time, thereby reducing the count of storms that may exist in one location over multiple time steps. We would therefore expect that only the longest lasting storms would likely be delineated with this approach, which we expect would almost exclusively result in synoptic-scale multi-day events. By instead downscaling the spatial resolution, we expect to improve the delineation of a range of storm types across microscale, mesoscale, and synoptic duration and extents. We would therefore expect to observe an equal improvement in model skill when reconstructing microscale and mesoscale storms that may produce anything from flash flooding to tropical storm impacts within the validation database.

Furthermore, the scope of this manuscript was to demonstrate the application of the temporal geostatistical method to hydrometeorological datasets. In a subsequent manuscript, we plan to employ a higher resolution precipitation dataset that will enable comparison between space time metrics.

- 1. Unclear connection to physical processes
- The discussion of seasonal variability in space-time metrics is insightful but a bit messy and extremely focused of heat waves in the current manuscript. For example, the biannual cycle in heat index metrics and annual cycle in precipitation metrics could be linked to known climatological characteristics. We agree with the reviewer and will expand this part of the manuscript to connect space time metrics to seasonal variability and regional to hemispheric processes that could explain these predictable patterns over eight decades. Indeed, Figure 5 already describes the seasonal cycle of space time metrics per variable. In particular, we will discuss the role of Sea Surface Temperature (SST) on relative humidity and the role of seasonal variability of surface temperature on heat index variability in the Southeastern US. Additionally, we will discuss annual role of Integrated Vapor Transport (IVT) on precipitable water availability that explains the initiation and termination of the water year, which may be correlated to the precipitation space time metric.
- The connection to physical processes can provide material supporting the
  robustness and usefulness of the method. I suggest creating a subsection
  dedicated to this topic in the discussion (it is now within the subsection on
  covariance modelling), clearly stating the meaning of the covariance results, and
  the connection to known physical processes, storm types and weather patterns.
   This is a great point that we plan to explore further in a new subsection describing

Space Time Metrics Inform Physical Processes. The content of this section is exactly what we propose in the above response.

- Under exploitation of long-term cluster creation: The long-term frequency of the
  created clusters tells something about the climatology of extreme events in the
  region since 1940. I see some results in the supplementary material, but they seem
  underexploited. Simple trends could be assessed on the number of clusters,
  average size, intensity. We agree with the reviewer and will incorporate this section
  of the SI into the main text.
- 1. Recommendations and Practical Implications
- The conclusions (Section 6) provide clear takeaways, but the recommendations for future research are somewhat generic. To make them more impactful, the authors should:
  - o Explicitly tie recommendations to the literature review. For example:
    - If previous studies (e.g., Tilloy et al., 2022; Liu et al., 2023) used biased clustering methods, how could the covariance-informed approach improve their results? We expect that clustering of ERA5 precipitation may result in undercount of events below spatial resolution of the dataset, resulting in an undercount of mesoscale storms. This may be less meaningful in Tilloy et al. 2022 since Extratropical Cyclones predominantly cause hazardous conditions in England, which we expect are well-resolved in ERA5.
    - Are there specific datasets or regions where this method would be most beneficial? We expect that the space time metric scales with different geophysical variables, including latitude, general circulation, large scale atmospheric organization (such as Rossby Waves and multi-decadal oscillations). We agree with both points and will address these upon revision.
  - o Address scalability: Could this framework be applied globally? What are computational challenges? Yes, this framework could be applied globally but is computationally restricted by the scaling of covariance modeling with spatial and temporal additions. Covariance modeling scales linearly with additional spatial locations but exponentially with temporal locations. As such, we expect that a global framework could be enabled with 1) large computational resources conditioned on 2) temporal windows adjusted to the available computational resources. Comparison of global with regional

space time metrics could provide consistent definition of the scale and persistence of regional meteorological processes that has not been available until now.

**Specific comments**

- Abstract (Line 10-15): The phrase "few studies test whether a dataset meets this
  requirement" could be more precise. For example: "While spatiotemporal clustering
  is widely used, few studies quantitatively assess whether a dataset's resolution
  satisfies the normalization assumption required for unbiased clustering." We agree
  and will make the recommended change.
- Line 146-158 p.6: It seems that you have two different uses of the acronym BME. The line on 158 does not offer an acronym but rather refers to a previously-defined acronym. We will update this line to be best unbiased nonlinear estimator (e.g., BME).
- Line 207 p.9: The space-time ratio was already used in Tilloy et al.,2022.
   Furthermore, why do you introduce this ratio if you don't use it? (Line 378). We agree with the author that a space-time ratio was applied in Tilloy et al., 2022. We use the same approach in this analysis but simply call the space-time ratio a space time metric as this terminology already exists in the field of temporal geostatistics from which the BME method was produced.
- Line 245 p.10: Formatting issue We agree and will revise this formatting issue upon resubmission.
- Line 254 p.11: "Southeastern populations are the most frequent hotspot" do you mean regions? We do mean regions and will revise accordingly.
- Line 363 p.15: Why only the last 4 years of NOAA storm events? What was different before 2019. As shown in Table 2, the recall is calculated since 2000 yet displays greater skill post-2019. We expect that this is due to improved quality of the validation dataset over time.
- Figure 6: The choice of the colour scale breaks is odd, please find a more interpretable scale. We will change to a blue (low) to red (high) recall color scale. We did not want this validation figure to conflict with the color palettes of blue and red space time metrics.